



# Rapid landslide identification using synthetic aperture radar amplitude change detection on the Google Earth Engine

Alexander L. Handwerger[1,2], Shannan Y. Jones[3], Mong-Han Huang[3], Pukar Amatya[4,5,6], Hannah R. Kerner[7], and Dalia B. Kirschbaum[6]

[1]Joint Institute for Regional Earth System Science and Engineering, University of California, Los Angeles, CA, USA
[2]Jet Propulsion Laboratory, California Institute of Technology, Pasadena, CA, USA
[3]Department of Geology, University of Maryland, College Park, MD, USA
[4]Universities Space Research Association, Columbia, MD, USA
[5]Goddard Earth Sciences Technology and Research, Columbia, MD, USA
[6]Hydrological Sciences Laboratory, NASA Goddard Space Flight Center, Greenbelt, MD, USA
[7]Department of Geography, University of Maryland, College Park, MD, USA

*Correspondence to*: Alexander L. Handwerger (alexander.handwerger@jpl.nasa.gov) and Mong-Han Huang (mhhuang@umd.edu)

**Abstract.** The rapid and accurate mapping of landslides is critical for emergency response, disaster mitigation, and improving
our understanding of where landslides occur. Satellite-based synthetic aperture radar (SAR) can be used to identify landslides, often within days after triggering events, because it penetrates clouds, operates day and night, and is regularly acquired worldwide. Although there are many landslide detection methods using SAR, most require downloading a large volume of data to a local system and specialized processing software and training. Here we present a SAR-based amplitude change detection approach designed for those without SAR expertise that uses multi-temporal stacks of freely available data from the
Copernicus Sentinel-1 satellites to identify landslides on Google Earth Engine (GEE). We provide strategies that can aid in rapid response and event inventory mapping. We test our GEE-based approach in a ~277 km² area in Hiroshima Prefecture, Japan where ~3,800 landslides were triggered by rainfall in July 2018. Our ability to detect landslides improves with the total number of SAR images acquired before and after the landslide event, by combining both ascending and descending acquisition geometry data, and by using topographic data to mask out flat areas unlikely to experience landslides. Importantly, our GEE
approach allows the broader hazards and landslide community to utilize these state-of-the-art remote sensing data.

## 1 Introduction

Rapid response to landslide events (and other natural hazards) is necessary to assess damages and save lives. This response effort includes ground-based teams of local residents, government officials and logistical coordinators, scientists, engineers, and more, all working together to identify critically damaged areas (e.g., Benz and Blum, 2019; Inter-Agency Standing
Committee, 2015). Yet, many response efforts are impeded by a lack of detailed information on the condition or location of damaged areas following large and widespread landslide events (Lacroix et al., 2018; Robinson et al., 2019). In addition to rapid response efforts, it is also important to construct accurate landslide inventories in the weeks to months following these



events (Froude and Petley, 2018; Roback et al., 2018; Williams et al., 2018). These detailed inventories are used to improve understanding of where landslides occur, to quantify erosion, and to look for areas where secondary hazards, such as outburst

flooding due to landslide dams, may be occurring (Kirschbaum et al., 2015; Kirschbaum and Stanley, 2018; Roback et al., 2018). It is therefore necessary to develop easy-to-use tools with freely available data that can be used to map the landslide extent and level of damage following catastrophic events.

Remote sensing techniques are commonly used to construct landslide inventories over large areas following catastrophic events (e.g., Bessette-Kirton et al., 2019; Roback et al., 2018). Satellite-based optical imagery (e.g., Landsat,

Sentinel-2, ASTER) provides high quality information for landslide mapping with medium resolution (10-30 m pixel size). Many studies have leveraged these data with manual (e.g., Harp and Jibson, 1996; Liao and Lee, 2000; Massey et al., 2020) and semi-automated/automated mapping techniques to identify landslides (e.g., Amatya et al., 2019; Ghorbanzadeh et al., 2019; Hölbling et al., 2015; Lu et al., 2019; Mondini et al., 2011, 2013; Stumpf and Kerle, 2011). While optical imagery provides high quality data, it is often limited in rapid response efforts because optical imagery requires daylight as well as

shadow- and cloud-free conditions for accurately identifying landslides. For instance, following the 2015 Mw 7.8 Gorkha earthquake, which triggered more than 25,000 landslides, persistent cloud cover prevented landslide mapping from satellite optical imagery for more than a week (Burrows et al., in review; Williams et al., 2018; Robinson et al., 2019).

Freely available satellite-based synthetic aperture radar (SAR) circumvents some of these issues because it can penetrate through clouds and operate day or night, but is still limited by geometric shadow and distortion. SAR data have been

used for more than a decade to investigate landslides (e.g., Colesanti and Wasowski, 2006; Hilley et al., 2004; Roering et al., 2009). However, most studies have focused on interferometric SAR (InSAR), which measures the radar phase change between two acquisitions in order to quantify ground surface deformation (Handwerger et al., 2019; Huang et al., 2017b; Intrieri et al., 2017; Schlögel et al., 2015). SAR radar amplitude- and coherence-based change detection can also be used to identify landslides, floods, and other types of natural hazards (Burrows et al., in review; DeVries et al., 2020; Jung and Yun, 2020;

Mondini et al., 2019; Rignot and Van Zyl, 1993; Tay et al., 2020; Yun et al., 2015). Changes in amplitude and coherence occur when there are changes in ground surface properties (e.g., reflectance, roughness, dielectric properties) before and after landslide events. Coherence-based change detection methods work well in urban areas because normally the coherence is high prior to an event, and there is a reduction in coherence from damages after an event. However, amplitude-based change detection methods outperform coherence-based methods in densely vegetated mountainous regions where landslides tend to

occur, because in these areas coherence is always low (i.e., no change), while the amplitude can change (Jung & Yun, 2020). Currently, amplitude-based change detection methods are under-utilized for landslide mapping following catastrophic events, primarily due to data access and the specialized processing and software required to analyze SAR data.

In this study, we developed easy-to-use tools for those without SAR expertise to identify landslides for rapid landslide identification that can aid in rapid response and inventory analyses. We define rapid response landslide detection as the period

of time within one week following a landslide event. Recent studies have also defined rapid response over a two-week time period following an event (e.g., Burrows et al., in review; Inter-Agency Standing Committee, 2015; Williams et al., 2018).



Our tools run entirely in Google Earth Engine (GEE), a free cloud-based online platform, using only freely available data (Gorelick et al., 2017). Recent studies have used GEE to identify floods (DeVries et al., 2020), investigate land cover (Huang et al., 2017a) and surface water change (Donchyts et al., 2016), monitor agriculture (Dong et al., 2016), and more. We used

SAR data provided by the Copernicus Sentinel-1 (S1) satellites and a digital elevation model (DEM) provided by the NASA Shuttle Radar Topography Mission (SRTM) (Farr et al., 2007) for the landslide detection analysis. Our methodology requires only the spatial coordinates of the area of interest (AOI) and the dates and duration of the catastrophic event. We tested our landslide mapping tools in a ~277 km$^2$ AOI by analyzing a widespread landslide event that occurred in Hiroshima Prefecture, Japan during 2018 (Fig. 1) (Hirota et al., 2019). This record-breaking rainfall event between June 28 and July 9, 2018 triggered

~3800 landslides and killed more than 100 people in our AOI. We tested the accuracy of our SAR-based landslide detection by making comparisons with the official landslide inventory provided by the Geospatial Information Authority of Japan (GSI). We performed a sensitivity analysis by varying the time span (i.e., number of images) of SAR data used before and after the event and by incorporating topographic slope- and curvature-based masks to remove regions where landslides are unlikely to occur. We provide recommended approaches that can aid in rapid response and event inventory mapping. Our work highlights

the utility of using SAR amplitude data to detect landslides following catastrophic events, which is necessary for rapid response and for generating landslide inventories. Importantly, our Google Earth Engine approach does not require prior SAR training or software so that the broader hazards communities can utilize these state-of-the-art remote sensing data.

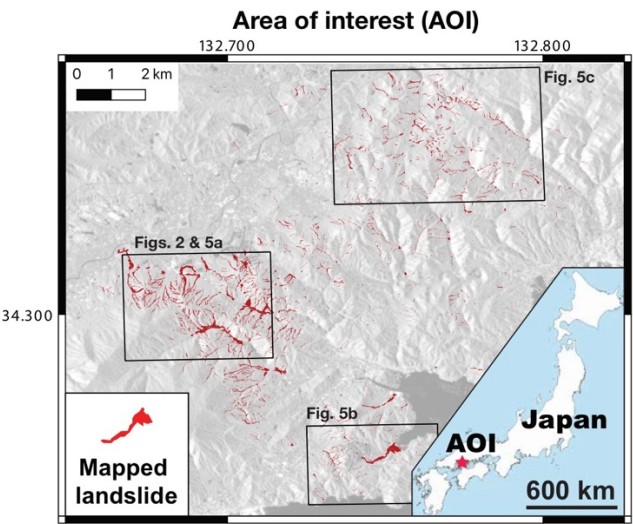


**Figure 1: Area of interest (AOI) showing a widespread landslide event that occurred in Hiroshima Prefecture, Japan during July 2018.** Red polygons show the GSI landslide inventory draped on Sentinel-1 amplitude stack. The amplitude stack consists of the median amplitude for all available Sentinel-1 images acquired between 2015 and 2020. Inset shows the AOI location within Japan. Black rectangles show the location of sub-areas highlighted in Figures 2 and 4.






## 2 Methods

The main goal of our work is to provide tools for those without SAR expertise that can be used to identify landslides as fast as possible after triggering events (i.e., rapid identification) and for constructing landslide inventories with high accuracy. Our methodology is developed in the GEE "playground" (browser-based graphical user interface) using the JavaScript application

programming interface (API) (Gorelick et al., 2017). This interface allows for coding, mapping/visualization, documentation, and more, and the products can be easily exported for offline analyses. The GEE code developed here is published on Github (see code availability).

### 2.1 SAR Amplitude-based Change Detection on Google Earth Engine

We use SAR amplitude data from the Copernicus Sentinel-1 (S1) satellite constellation. The S1 constellation currently consists

of two satellites, S1-A and S1-B, launched in March 2014 and April 2016, respectively. Each satellite has a minimum 12-day revisit time for a given area. Using data from both satellites provides a minimum 6-day revisit time. The S1 satellites carry a C-band radar sensor with wavelength ~5.6 cm. Depending on the location of the AOI, both ascending (asc) and descending (desc) S1 data may be available (see worldwide acquisition coverage, https://sentinel.esa.int/web/sentinel/missions/sentinel-1/observation-scenario). GEE can access S1 Ground Range Detected (GRD) products that are processed to remove thermal

noise and are radiometric and terrain calibrated using the SRTM DEM, or the Advanced Spaceborne Thermal Emission and Reflection Radiometer (ASTER) DEM for areas above 60 degrees latitude. The GEE S1 GRD collection is updated daily and new data are uploaded to GEE within two days after they become available. GEE ingests all of the available ascending and/or descending images on-the-fly. The spatial resolution of the Interferometric Wide Swath (IW) GRD products is 20 x 22 m, and the images have 10, 25, or 40 m pixel spacing and up to four polarization modes: 1) vertical transmit/vertical receive (VV), 2)

horizontal transmit/horizontal receive (HH), 3) vertical transmit/horizontal receive (VV + VH), and 4) horizontal transmit/vertical receive (HH + HV). All S1 GRD amplitude values are provided in logarithmic units of decibels (dB) calculated as $10*\log_{10}(A)$, where $A$ is the SAR amplitude. For this study we only used SAR data in the VH polarization. Cross-polarizations VH and HV are sensitive to forest biomass structure (Le Toan et al., 1992) and are therefore useful to identify landslides in vegetated areas. We encourage users to explore the use of other polarizations in future studies.

115        We begin by selecting an Area of Interest (AOI) and time period before ($T_{pre}$) and after ($T_{post}$) the Event of Interest (EOI). The AOI can be a single landslide or a mountain range, and the EOI can last from seconds (e.g., earthquakes) to several days (e.g., storms). Thus, it is important to consider the spatial and temporal scale of the EOI when selecting the AOI, $T_{pre}$, and $T_{post}$. Furthermore, $T_{pre}$ and $T_{post}$ will vary depending on the goal of the project (i.e., rapid identification or inventory construction). To reduce noise from poor quality data, we removed all pixels with values ≤ -30 dB (based on a recommendation

from the GEE S1 Data Catalog). We further reduced noise by stacking images to create pre-event and post-event stacks. Each stack was calculated as the temporal median of the data. We constructed image stacks using ascending data, descending data,





and combined ascending + descending data. The combined ascending + descending data (also referred to as asc + desc) were calculated as the mean of the ascending and descending stacks.

We identified potential landslides (and other ground surface change) by examining the change in amplitude, $A_{ratio}$,

defined as the pre-event stack minus post-event stack, $A_{pre} - A_{post}$. Because GEE provides the amplitude data in dB, $A_{pre} - A_{post}$ is equivalent to the standard amplitude ratio approach $A_{ratio} = \log_{10}(A_{pre}/A_{post})$, which is the most commonly used amplitude-based change detection method (e.g., Jung and Yun, 2020; Mondini et al., 2019). $A_{ratio}$ can be either positive or negative, with positive values corresponding to a decrease in SAR amplitude after a landslide event. The SAR amplitude changes following landslide events because landslides cause major changes in ground surface properties that alter the radar reflectance, hillslope

geometry, roughness, and dielectric properties (Adriano et al., 2020; Rignot and Van Zyl, 1993).

Lastly, to reduce false positives, we removed areas that are unlikely to correspond to landslides (e.g., oceans, lakes, flat surfaces, hilltops) by using threshold-based masks made from topographic slope and curvature calculated from the 1 arc-second (~30 m) resolution SRTM DEM. The SRTM DEM and the topographic slope products are available in GEE (Fig. 2b), and we calculated the curvature by taking the Laplacian of topography with a Gaussian spatial filter (standard deviation = 60

m and radius = 120 m) as part of our processing in GEE (Fig. 2c). We refer to this mask throughout this paper as the "DEM mask".

## 2.1 SAR Change Detection Performance

To test the performance of our SAR amplitude change detection for landslides, we compared our results to the inventory provided by the GSI for our ~277 km$^2$ AOI. We removed landslides with area < 100 m2 from the GSI inventory for

a better match with the minimum size of a 10 m x 10 m S1 SAR pixel size in GEE. After removing the smallest landslides, 3,370 individually mapped polygons remained in the GSI inventory. We calculated detection performance using Receiver Operating Characteristic curves (ROC) (Fan et al., 2006), which measure our detection compared to the GSI inventory under a variety of thresholds for discriminating between landslide and non-landslide pixels. We computed the ROC curves for all pixels within the entire ~277 km$^2$ AOI shown in Figure 1. For these analyses, each pixel in the SAR amplitude change raster

was classified as a landslide if the $A_{ratio}$ pixel value was greater than a threshold value, or non-landslide if it was less than the threshold value. We then compared these classified pixels to the true landslides in the GSI inventory. The ROC curve is calculated by varying the $A_{ratio}$ threshold values used to classify each pixel as a landslide or a non-landslide. The initial $A_{ratio}$ threshold is set as the minimum value in the dataset and is increased until reaching the maximum $A_{ratio}$ value. For each threshold, the false positive rate, defined as the ratio of false positives to true non-landslide pixels, is compared to the true

positive rate, defined as the ratio of true positives to true landslide pixels. The best performance is determined by maximizing the Area Under the ROC Curve (AUC) (Fan et al., 2006). An AUC = 1 corresponds to a perfect classifier while an AUC = 0.5 is equivalent to a random selection (50% true positive rate and 50% false positive rate). To maximize the AUC, we performed a sensitivity analysis by varying the pre-event and post-event time periods, the satellite acquisition geometry (i.e., ascending, descending, or ascending + descending), and the thresholds used for slope angle and curvature for the DEM mask. We also



converted the SAR-based amplitude change detection pixels to a binary raster image that includes values of 1 at possible landslide locations and values of 0 at possible non-landslide locations. This provides us with a pixel-based map that can be converted to a Kernel Density Estimation "heat map" the QGIS software package. The heat map is a density raster made from point vectors by calculating the density of points in a location. Heat maps can be used to highlight areas with high landslide density.

## 3 Test Site

A record-breaking rainfall event occurred between 28 June and 8 July, 2018 in west and central Japan that resulted in widespread floods and landslides. There were more than 200 fatalities, 20,000 damaged buildings and 8,500 damaged houses caused by these natural disasters. Hiroshima Prefecture had an especially high 108 fatalities, 14,862 damaged buildings, and 689 destroyed houses, significantly more than other Prefectures, primarily due to the ~8000 triggered landslides (Adriano et
al., 2020; Hirota et al., 2019; Miura, 2019). Between 1–7 July, 2018 there was approximately 500 mm of rainfall in the Hiroshima Prefecture.

The mountains of the Hiroshima Prefecture are underlain by weathered granite and covered with dense forest (Hirota et al., 2019; Miura, 2019). The change in land cover from dense forest to landslide scars and deposits causes changes in surface reflectance and hillslope geometry that is detectable by measuring the change in SAR amplitude (e.g., Adriano et al., 2020).
Our study site is a ~277 km² region in the Hiroshima Prefecture that had approximately ~3800 landslides mapped by the GSI (Fig. 1). Our AOI has a mixture of land cover including dense forest with little infrastructure in the mountains and farmlands and residential areas and cities in the valleys. The minimum, mean, and maximum elevation is 0 (sea level), 210, and 850 m respectively. The mean slope angle is 14° ± 11° (± 1 standard deviation) with a maximum slope of 64°.

## 4 Results

### 4.1 Landslide identification

Our SAR-based amplitude change detection for landslide identification shows areas that correspond to the landslides mapped by GSI and are also observed with the Sentinel-2 optical imagery that is also available in GEE (Fig. 2). The quality of the SAR-based landslide detection depends on the total number of images used in the pre-event and post-event stacks, slope and curvature threshold, $A_{ratio}$ threshold, and whether we use ascending, descending, or combined ascending + descending data. To
determine the best routine for landslide identification, we compared our SAR-based amplitude change detection with the GSI inventory using the AUC scores computed from the ROC curves (Fig. 3 and Table 1). The full list of SAR data used in our analyses are in Tables S1 and S2. First, we used all of the SAR data (ascending, descending, and ascending + descending data) that was available on the starting day of our analysis for this study (May 29, 2020). The pre-event stack consisted of 141 images (100 ascending and 41 descending) with the first image collected 1,152 days before the EOI and the last image collected


12 days before the EOI. The post-event stack consisted of 136 images (78 ascending and 58 descending) with the first image collected 1 day after the EOI and the last image collected 684 days after the EOI. Figure 2 shows a sub-area of our AOI with many of the mapped landslides detected by our SAR-based amplitude change approach. We found that these landslides primarily caused a positive $A_{ratio}$ (i.e., post-event amplitude < pre-event amplitude), but there were also some negative amplitude changes within landslide scars as discussed in Adriano et al. (2020). We also found that ascending and descending

satellite tracks each show many landslides, but the best approach is to combine ascending + descending images into a single stack. We found AUC scores of 0.7363, 0.7409, 0.7712 for ascending, descending, and ascending + descending, respectively using the complete pre- and post-event stacks. We repeated these analyses by changing the time duration for images included in $T_{pre}$ and $T_{post}$ to 12, 6, 3, and 1 months, as well as using only a single image before and after the EOI (Figs. 3a and A1, and Table 1). The highest AUC occurred when using all available pre-event and post-event data and by combining ascending +

descending data.

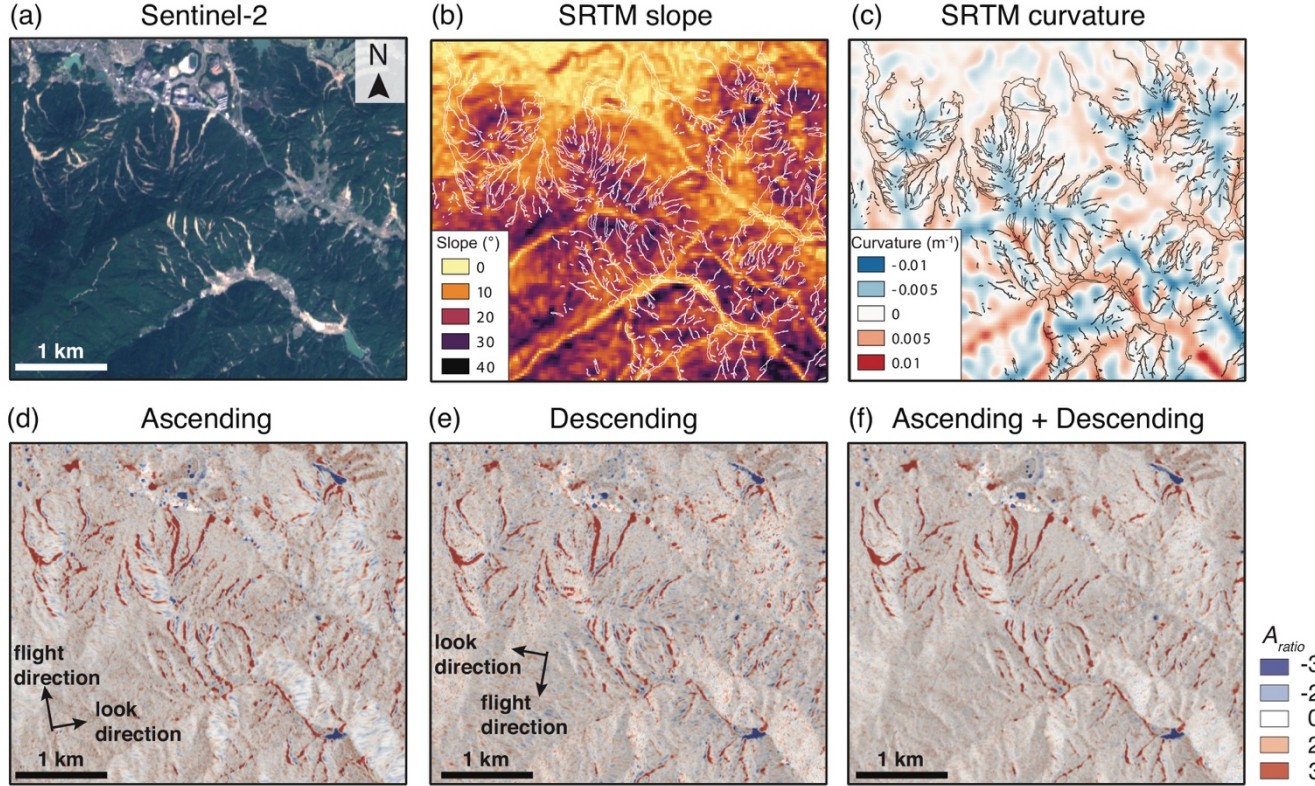

**Figure 2: Landslides triggered by the 2018 rainfall event for a sub-area of our Hiroshima Prefecture AOI.** (a) Sentinel-2 optical image showing landslide scars. (b) Topographic slope and (c) curvature map with white and black polygons, respectively, showing landslide

inventory mapped by the GSI. Both slope and curvatures are calculated using the ~30 m SRTM DEM. (d,e,f) SAR amplitude change for ascending, descending, and ascending + descending satellite acquisitions. These maps were created using all available pre-event and post-event SAR data and represent our best case landslide identification (see Index 1 in Tables 1, S1, and S2). Red colors show positive values that correspond to decrease in SAR amplitude following landslide events. Most landslides cause a decrease in SAR amplitude, however we note there are also some increases in amplitude within the landslide zone. See location in Figure 1.

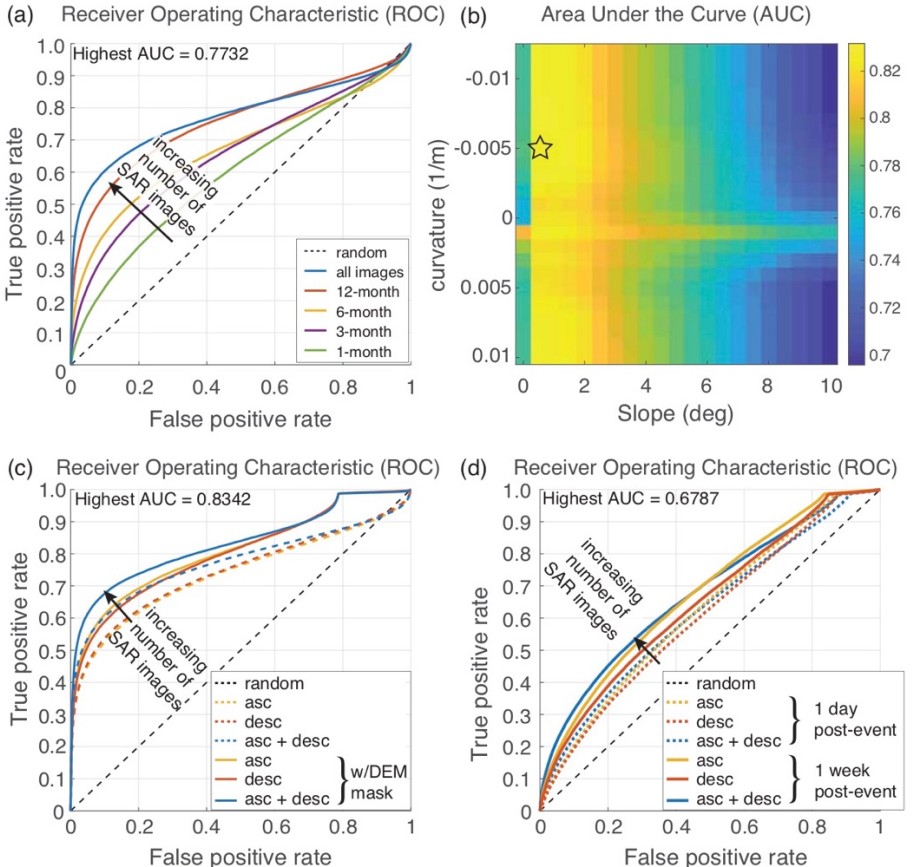


**Figure 3: Receiver Operating Characteristic (ROC) and Area Under the Curve (AUC) analyses to determine the best approach for landslide identification.** (a) Colored lines show ROC curves as a function of pre-event stack time period ($T_{pre}$) and post-event stack time period ($T_{post}$) (see Tables 1, S1, and S2). (b) AUC score parameter space with topographic slope and curvature thresholds calculated using all available pre-event and post-event images (highest AUC in (a)). Black star shows the threshold that maximizes the AUC by masking out

slopes < 0.5 degrees and topographic curvature > -0.005 m-1. (c) ROC curves using all available pre- and post-event data (highest AUC in (a)). Dashed colored lines correspond to ascending (asc), descending (desc), and ascending + descending data. Solid colored lines show the same data with the addition of the DEM mask slope and curvature thresholds. (d) ROC curves with DEM mask for rapid response mapping using all pre-event SAR data and post-event data acquired 1 day and 1 week following the landslide event.

Next we used topographic data to help reduce false positives and increase the AUC score (Figs. 3 and A1). To

determine the slope and curvature thresholds, we used our best landslide identification approach from the previous analyses

(i.e., all available pre-event and post-event data) and found the thresholds that maximized the AUC. We found that we can

further improve the AUC to 0.8101, 0.8041, 0.8342 for ascending, descending, and ascending + descending, respectively, by

excluding areas with topographic slope < 0.5° and curvature > -0.005 m⁻¹. These areas of low slope correspond to flat regions,

such as cities and valley bottoms, and areas of relatively high positive curvature correspond to hilltops, where landslides are

less likely to occur (Figs. 3b and 3c). These same thresholds can be used for landslide detection in future events, but may need

to be fine-tuned empirically since future events will not have a landslide inventory available to maximize AUC scores against.



Table 1 shows the AUC scores for our analyses as a function of $T_{pre}$ and $T_{post}$. We find that AUC decreases with a decreasing number of images used in each stack. For instance, the AUC for ascending + descending using all available data was 0.8342. If we reduce $T_{pre}$ and $T_{post}$ to only include images from 1 year before and after the event, the AUC score drops to

0.8194. The AUC score decreases monotonically as the time span of images before and after the event is shortened.

| Goal | Index | Pre-event stack (number of images used) | Post-event stack (number of images used) | AUC score without DEM mask (asc, desc, asc+ desc) | AUC score with DEM mask (asc, desc, asc + desc) |
|---|---|---|---|---|---|
| Event inventory mapping | 1 | All available pre-event images (100 asc + 41 desc = 141 images) | All available post-event images (78 asc + 58 desc = 136 images) | 0.7376, 0.7435, 0.7732 | 0.8101, 0.8041, 0.8342 |
| Rapid Response | 2 | All available pre-event images (100 asc + 41 desc = 141 images) | First available post-event images (1 asc + 1 desc = 2 images) | 0.5617, 0.5463, 0.5825 | 0.6245, 0.6079, 0.6212 |
| Event inventory mapping | 3 | All images within 1 year pre-event (61 asc + 30 desc = 91 images) | All images within 1 year post-event (25 asc + 32 desc = 57 images) | 0.7043, 0.7262, 0.7550 | 0.7800, 0.7918, 0.8194 |
| Rapid Response | 4 | All images within 1 year pre-event (61 asc + 30 desc = 91 images) | First available post-event images (1 asc + 1 desc = 2 images) | 0.5639, 0.5510, 0.5869 | 0.6260, 0.6116, 0.6249 |
| Event inventory mapping | 5 | All available images within 6 months pre-event (29 asc + 15 desc = 44 images) | All available within 6 months post-event (14 asc + 17 desc = 31 images) | 0.6560 0.6218, 0.6722 | 0.7374, 0.7193, 0.7566 |
| Rapid Response | 6 | All available images within 6 months pre-event (29 asc + 15 desc = 44 images) | First available post-event images (1 asc + 1 desc = 2 images) | 0.5575, 0.5420, 0.5763 | 0.6200, 0.6042, 0.6153 |
| Event inventory mapping | 7 | All available images within 3 months pre-event (14 asc + 7 desc = 21 images) | All available within 3 months post-event (7 asc + 9 desc = 16 images) | 0.6207, 0.6204, 0.6561 | 0.6935, 0.7102, 0.7248 |



| Rapid Response | 8 | All available images within 3 months pre-event (14 asc + 7 desc = 21 images) | First available post-event images (1 asc + 1 desc = 2 images) | 0.5511, 0.5478, 0.5732 | 0.6110, 0.6050, 0.6098 |
|---|---|---|---|---|---|
| Event inventory mapping | 9 | All available images within 1 month pre-event (5 asc + 2 desc = 7 images) | All available within 1 month post-event (3 asc + 4 desc = 7 images) | 0.5626, 0.5787, z | 0.6142, 0.6415, 0.6350 |
| Rapid Response | 10 | All available images within 1 month pre-event (5 asc + 2 desc = 7 images) | First available post-event images (1 asc + 1 desc = 2 images) | 0.5338, 0.5324, 0.5513 | 0.5782, 0.5858, 0.5810 |
| Rapid Response | 11 | 1 asc +1 desc pre-event = 2 images | 1 asc +1 desc pre-event = 2 images | 0.5364, 0.5359, 0.5559 | 0.5624, 0.5833, 0.5770 |

**Table 1.** AUC scores for different pre- and post-event stacks. Goal column indicates whether the analyses were designed for rapid response or event inventory mapping. Index corresponds to the numbers listed in the legend in Figure A1. Complete list of SAR data used in each stack are shown in Tables S1 and S2 with indexed tabs.

### 4.2 Landslide identification for rapid response

To identify landslides for rapid response (i.e., within one week following the landslide event) we applied the DEM mask and explored landslide detection scenarios where we have limited post-event data. Ideally, for rapid response, the first or first few available images following a catastrophic event will provide key information to identify damaged areas. Thus, our methodology was designed with the goal of being able to provide information to responders on the location of critically damaged areas as quickly as possible. For rapid response, we quantified the change in SAR amplitude for a stack consisting of all of the available pre-event imagery (best approach for landslide identification in Sect. 4.1) and post-event imagery collected within 1 week of the EOI (Table 2). As mentioned above, the first available S1 image was acquired 1,152 days before the EOI and the last image was acquired 12 days before the EOI (Tables S1 and S2). The first post-event images were acquired on July 10, 2018 on both ascending and descending tracks, less than 1 day after the rainfall ended. Using only these 2 post-event images, we found an AUC of 0.6212 (Figs. 3d, 4 and Table 2). The second set of post-event images were acquired on July 16, 2018 on both ascending and descending tracks. Incorporating 4 total post-event images increased the AUC to 0.6787 (Figs. 3d, 4 and Table 2). Importantly, the AUC improves rapidly over the first week with the transition from 2 to 4 post-event images, indicating that the images immediately following the event provide key information on the location of damages for rapid response.



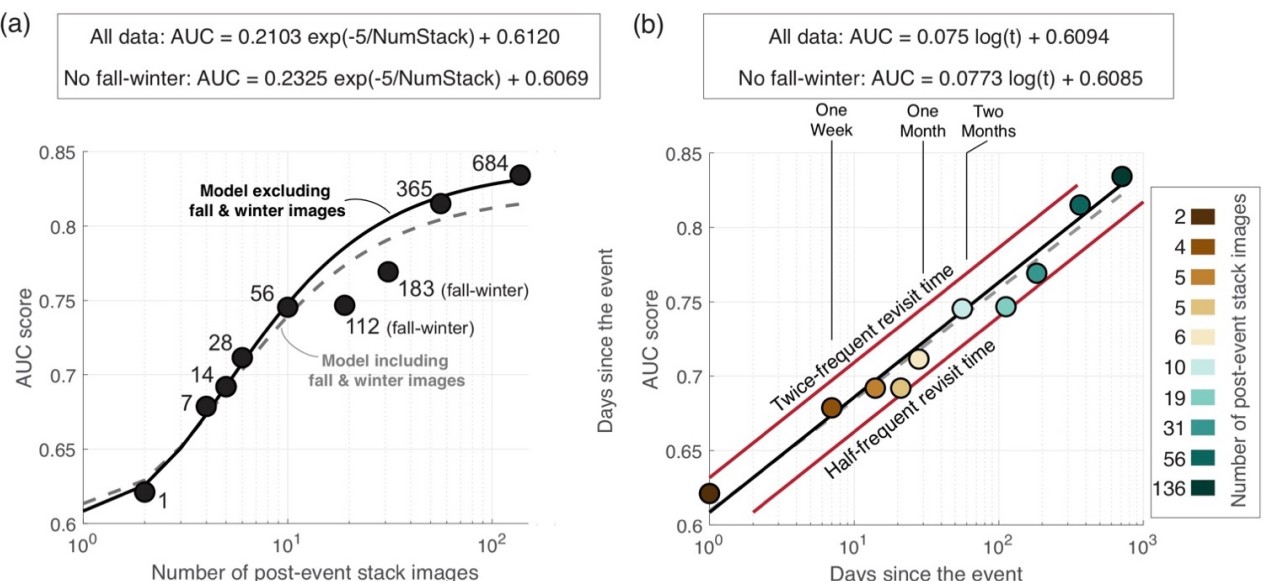

**Figure 4: Area Under the Curve (AUC) score as a function of increasing number of days and SAR images following the landslide event**. (a) AUC score as a function of number of SAR images used in the post-event stack. Black dashed and solid lines show exponential fit models with and without fall-winter SAR images, respectively, where NumStack is the number of post-event images used in the stack. The rate of AUC increase decays exponentially with the number of images in the stack. Numbers next to black circles correspond to the number of days since the landslide event. The two fall-winter SAR images labeled fall off the best-fit line, which we infer is due to seasonal differences in SAR amplitude (see Sect. 5). (b) AUC score as a function of days since the landslide triggering event. The dashed line shows the AUC score fit model and the black line shows a model fit excluding fall-winter images. The red lines indicate the predicted AUC score time series by assuming a satellite revisit time that is twice or half the S1 satellites for this region. Color scale corresponds to the number of images used in the post-event stack.

We continued to increase the $T_{post}$ duration following the rainfall event to determine when we would achieve the maximum AUC (Figs. 4 and A1). We found that the AUC continues to increase with post-event time, and thus the number of images acquired. The AUC is > 0.7 in just 28 days (6 post-event SAR images) after the event and reaches a maximum of 0.8342 at 684 days (136 post-event images) after the event. This increase in AUC can be approximated as an increasing form of exponential decay, with a rapid increase in AUC shortly after the event followed by a slower increase in AUC several months after the event (Figs. 4a and A3). Using our model fits, we also simulated the AUC score for hypothetical changes in the satellite revisit time. We found that if the satellite revisit was twice the current revisit time, the modeled AUC score would be ~0.7 just 1 week after the EOI, while if the revisit time was half the current revisit the modeled AUC score would be ~0.65.

While the AUC scores are relatively low for the rapid response analyses (Figs. 3 and. 4), the SAR-based amplitude change still provides key information that can be used to identify the critically damaged areas. To highlight these areas we employed a "heat map" approach (Fig. 5). The heat map is an interpolation technique that highlights areas with spatially dense amplitude change values that are classified as landslides based on our ROC analysis. To produce the heat map, we set an

amplitude threshold that corresponds to a true positive rate of 0.3 and false positive rate of 0.1 in the ROC curve, which removes $A_{ratio} < 1$. We further removed isolated pixels when there are less than 6 neighboring pixels, which we attributed to noise rather than true landslides. Figure 5 shows the GSI landslide inventory and the corresponding heat maps for the two rapid response scenarios (2 post-event images and 4 post-event images) for three different sub-areas in our AOI. The heat maps highlight areas that were critically damaged by landslides using images acquired 1 and 6 days after the rainfall event. Many of

the largest landslides are clearly visible with numerous smaller landslides hidden by noise. The heat maps using 4 post-event images significantly improved the landslide detection (also shown by the AUC) because the additional 2 images helped reduce the noise. We note that many of the false positives appear to occur in farmlands that are near the mountain front.

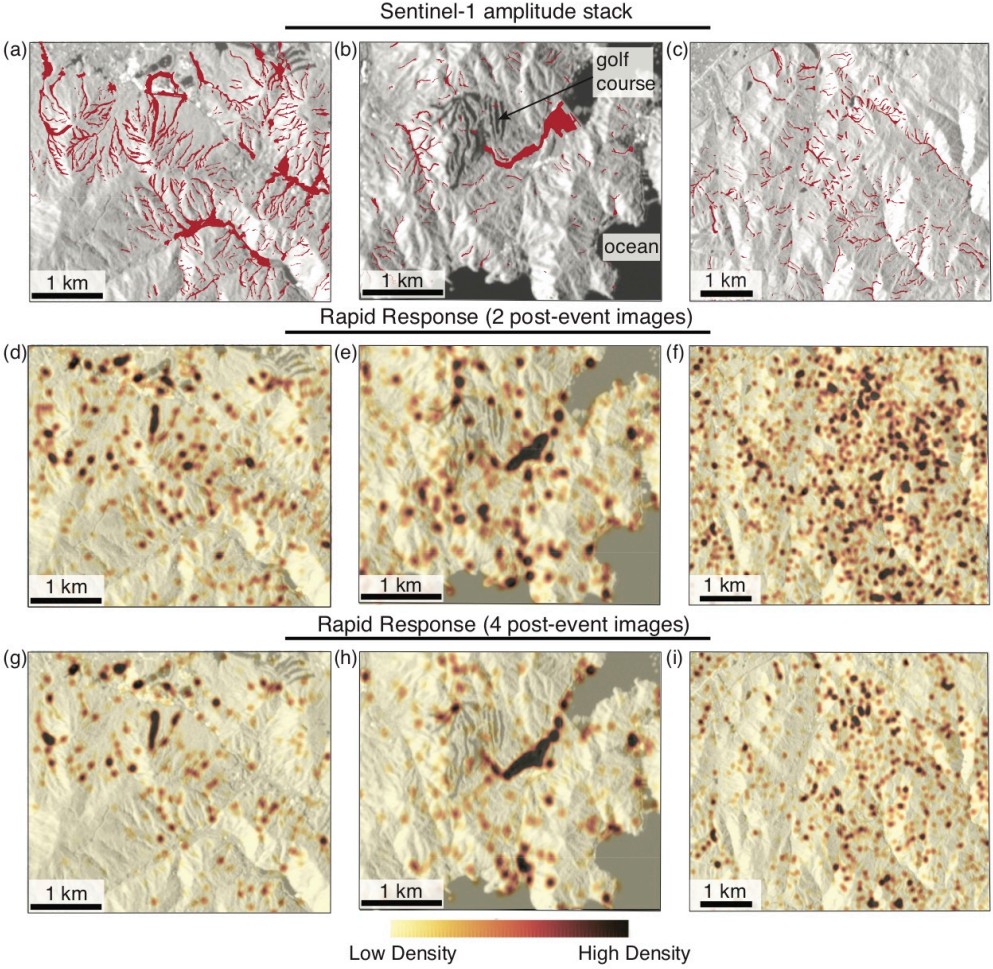

**Figure 5: Rapid response heat maps for three sub-areas showing a variety of different landslide sizes.** Locations of each sub-area are
shown with black rectangles in Figure 1. (a–c) SAR-amplitude stack with GSI landslide polygons shown in red. (d–f) Rapid response heat maps using two post-event images (one ascending and one descending) that were acquired less than one day after the EOI (July 10, 2018) draped on SAR-amplitude stack. (e–f) Rapid response heat maps with four post-event images (two ascending and two descending) that were acquired within 6 days after the EOI on July 10, 2018 and July 16, 2018) draped on the SAR amplitude stack.





| Index | Pre-event stack (number of images used) | Post-event stack (number of images used) | AUC score without DEM mask (asc + desc) | AUC sore with DEM mask (asc + desc) | Final acquisition date used in stack |
|---|---|---|---|---|---|
| 12 | All available pre-event images (100 asc + 41 desc = 141 images) | First available post-event images (1 asc + 1 desc = 2 images) | 0.5835 | 0.6212 | 2018-07-10 |
| 13 | All available pre-event images (100 asc + 41 desc = 141 images) | All available within 1 week post-event (2 asc + 2 desc = 4 images) | 0.6275 | 0.6787 | 2018-07-16 |
| 14 | All available pre-event images (100 asc + 41 desc = 141 images) | All available within 2 weeks post-event (2 asc + 3 desc = 5 images) | 0.6400 | 0.6919 | 2018-07-23 |
| 15 | All available pre-event images (100 asc + 41 desc = 141 images) | All available within 3 weeks post-event (2 asc + 3 desc = 5 images) | 0.6400 | 0.6919 | 2018-07-30 |
| 16 | All available pre-event images (100 asc + 41 desc = 141 images) | All available within 4 weeks post-event (2 asc + 4 desc = 6 images) | 0.6542 | 0.7114 | 2018-08-06 |
| 17 | All available pre-event images (100 asc + 41 desc = 141 images) | All available within 8 weeks post-event (4 asc + 6 desc = 10 images) | 0.6760 | 0.7454 | 2018-09-03 |
| 18 | All available pre-event images (100 asc + 41 desc = 141 images) | All available within 16 weeks post-event (8 asc + 11 desc = 19 images) | 0.6710 | 0.7467 | 2018-10-29 |
| 19 | All available pre-event images (100 asc + 41 desc = 141 images) | All available within 6 months post-event (14 asc + 17 desc = 31 images) | 0.6854 | 0.7693 | 2019-01-09 |



| 20 | All available pre-event images (100 asc + 41 desc = 141 images) | All images within 1 year post-event (24 asc + 32 desc = 56 images) | 0.7477 | 0.8151 | 2019-07-09 |
| 21 | All available pre-event images (100 asc + 41 desc = 141 images) | All available post-event images (78 asc + 58 desc = 136 images) | 0.7732 | 0.8342 | 2020-06-01 |


**Table 2.** AUC scores for different post-event stacks. Complete list of SAR data used in each stack are shown in Tables S1 and S2. Index corresponds to the indexed tabs in Tables S1 and S2.

**5 Discussion**

Our results show that SAR-based amplitude change in GEE can be used to detect landslides over large areas. Our main goal was to develop an easy-to-use methodology for those without SAR expertise that can be used to 1) construct an accurate landslide inventory and 2) create landslide density maps (i.e., heat maps) that can aid in rapid response to catastrophic landslide events. We have demonstrated that rainfall-triggered landslides in the Hiroshima Prefecture cause an overall decrease in SAR amplitude. This decrease in SAR amplitude occurs because the landslide scar and damage acts to decrease backscattering

reflectance to the satellite relative to a pre-failure ground surface (Adriano et al., 2020). We also observed numerous places where SAR amplitude change is not related to landslides (Figs. 2 and 5). In many cases, these amplitude changes occur in farmlands where the ground surface is commonly worked and there can be large changes in water content due to flooding, which influences the SAR reflectivity (Adriano et al., 2020; DeVries et al., 2020; Jung and Yun, 2020). These false positives are particularly noticeable in the rapid response imagery (Fig. 5).

We found that increasing the total number of SAR images used in the SAR amplitude stacks improved landslide detection, and that the images immediately following the EOI provide the greatest increase in detection performance (Fig. 4). The detection performance was further improved by applying a DEM mask to remove areas where landslides are unlikely to occur. Our highest AUC score for landslide inventory mapping included all available SAR images before and after the EOI and our highest AUC for rapid response used all pre-event images and 4 post-event images acquired 6 days after the EOI (Fig.

3 and Tables 1 and 2). Importantly, the stack of two post-event images (acquired less than 1 day after the EOI) also provides a useful landslide heat map for a rapid response (Fig. 5). We also found that the AUC continued to increase with each image acquisition following the EOI (Fig. 4). Although the AUC score increases following the EOI, the rate of AUC increase is temporarily reduced between 56 and 183 days after the event. We infer that this change in the rate of AUC increase is due to seasonal changes in vegetation. The image stacks made between 56 and 183 days after the event contain a relatively high

number of images collected in fall and winter (between November 2018 and February 2019) when vegetation cover is likely



reduced relative to the average yearly vegetation cover represented in the long-term pre-event stack. The change in the rate of AUC increase is clearly shown by examining the fitted lines in Figure 4b. The rate of the AUC score increase returns back to the overall trend after the spring season, possibly due to the growth of vegetation.

315        The increased number of SAR images used in pre- and post-event stacks helps to reduce noise from radar scattering and speckle in the SAR data. Recent work by Burrows et al. (in review), based on recommendations from the Inter-Agency Standing Committee (2015), defined the rapid response time period as two weeks following an event. For our dataset, only one additional S1 image was acquired in the second week after the EOI (Fig. 4). This additional S1 image resulted in an AUC increase from 0.6781 to 0.6911 and further highlights the sometimes inconsistent image collection from Earth observing satellites. For most regions around the world, ascending and descending S1 images are collected roughly 3 days apart, which
means it should take no more than 12 days to acquire at least 4 post-event images (2 ascending and 2 descending). However, the European Space Agency (ESA) does not store all S1 images on every flight path and also does not always acquire both ascending and descending data, which may hinder rapid response efforts in certain cases. Future work will explore noise reduction techniques such as image smoothing filters, which could be used in cases where the total number of available images is low. Our GEE code that accompanies this paper includes an option for implementing a Gaussian smoothing image filter (but
is not sufficiently tested in this study).

        We found that combining ascending + descending geometry SAR data into a single stack together improved landslide detection when compared to using ascending or descending data individually (Fig. 3c and Fig. 3d). Combining ascending + descending data into a single stack helps reduce bias introduced from the acquisition geometry (e.g., radar shadows, foreshortening, layover) (Adriano et al., 2020). The combined effect of stacking hundreds of images with both geometries is a
major improvement from previous SAR-based amplitude change studies that have focused on individual acquisition geometries and a relatively small number of SAR images (Adriano et al., 2020; Jung and Yun, 2020; Mondini et al., 2019). Our findings indicate that future catastrophic events will benefit from a large number of pre-event images (S1 data has been collected since 2014). Our findings also show that multitemporal SAR amplitude change detection works best for satellites with high repeat acquisition frequency (e.g., Fig. 4b).

335        While our analysis used the full resolution (10 m pixel spacing) S1 GRD data to identify landslides, several recent studies have applied a landslide density map approach to identify damaged areas, rather than focus on the location of individual landslides. Landslide density maps are made by counting the number of landslides within a fixed area. Bessette-Kirton et al. (2019) used a 2 x 2 km grid with optical satellite imagery and assigned high landslide density for > 25 landslides, low landslide density for 1–25 landslides, or no landslides for landslides triggered during Hurricane Maria in Puerto Rico, USA. Burrows et
al. (in review) used SAR coherence change with S1 and ALOS-2 data at ~200 x 220 m resolution to generate coherence change density maps for landslides triggered by the 2015 Mw 7.8 Gorkha, Nepal earthquake, the 2018 Mw 6.6 Hokkaido, Japan earthquake, and two 2018 Mw 6.8 and 6.9 Lombok, Indonesia earthquakes. These density maps provide useful information that can be used to identify critically damaged areas for rapid response. Our heat map approach (Fig. 5) is similar to a density





map. Both methods can be employed for rapid response and will help reduce issues related to noise when there are few available
post-event images.

We only explored SAR amplitude change with the C-band Sentinel-1 satellites because these are the only SAR data freely available in GEE. However, our methodology can also be applied to SAR satellites operating with different radar wavelengths and different pixel resolutions (e.g. X- and L-bands). Recent work by Adriano et al. (2020) used SAR amplitude change detection with data from the JAXA ALOS-2 satellite to identify landslides for the same EOI in the Hiroshima
Prefecture. ALOS-2 PALSAR-2 data has a L-band radar (~24 cm radar wavelength), which better penetrates through dense vegetation. They were able to clearly detect many landslides in their study area. Unfortunately, we were not able to make a direct comparison with their dataset due to issues related to data availability. Jung and Yun (2019) also used ALOS-2 data with SAR amplitude change, as well as coherence change methods to detect landslides after the 2018 Mw 6.6 Hokkaido earthquake. They found that a multitemporal amplitude correlation method provided the best landslide detection in the
vegetated mountains of Japan. Unfortunately, ALOS-2 is collected relatively infrequently worldwide and is not freely available which limits the use of our multitemporal and open source SAR amplitude change approach with these data. The NASA-ISRO SAR (NISAR) mission, which is currently expected to launch in January 2022, will operate with an L-band (~24 cm) SAR sensor and is designed to fly by the same location every 12 days. As L-band data can generally produce improved results in vegetated regions (Yun et al., 2015; Jung and Yun, 2019; Burrows et al., in review), we expect an improvement in our multi-
temporal stacking SAR amplitude change approach to detecting natural hazards using images from the NISAR mission. Similar to the Sentinel program the NISAR products will be publicly available. If GEE also ingests the NISAR products, the same GEE scripts provided in this study can be used for the NISAR images.

Although we were able to successfully identify many landslides with high accuracy, we expect that SAR amplitude change detection may require different processing strategies in different environments. For example, landslides that occur in
regions with different types of land cover may result in a different range of $A_{ratio}$ values, while regions that have significant seasonal changes (e.g., snowfall, vegetation cover) may require all pre-event and post-event data to be from the same season. Also, SAR data collection is not the same in all places around the world. For areas that have more frequent S1 data collection, we expect better ability to rapidly identify landslides, while the opposite is true for regions with less data collection (Fig. 4b). For our future work, we will use our GEE approach to explore how our multi-temporal amplitude change stacking detection
methods perform in different environments and in different climates given each will have different amounts of available S1 data. Nonetheless, our methodology presented throughout this manuscript, including stacking approaches and $A_{ratio}$ values provide a good starting point for all landslide events worldwide. Preliminary tests for different landslide events around the world, including the 2018 rainfall-triggered landslides in Montecito, CA, USA, the 2018 Mw 7.5 Papua New Guinea earthquake, and the 2018 earthquake-triggered landslides in Hokkaido, Japan, shows promising results using our method to
quickly identify many landslides in different environments.

## 6 Conclusions



In this manuscript, we developed a new methodology to detect landslides (and other ground surface changes) using freely available Sentinel-1 SAR data, topographic data, and open source tools in the Google Earth Engine platform. Our approach is

novel because it does not require specialized SAR processing software or training, and furthermore, it does not require the user to download any data to a local system. We found that the ratio of two multi-temporal SAR amplitude image stacks, composed of pre- and post-landslide event data, can be examined to identify catastrophic landslides for rapid response (within one week of a landslide event) and constructing accurate landslide inventories. The best landslide event inventory mapping approach combined all available SAR images acquired on ascending and descending satellite flight paths with topographic data to mask

out areas that were unlikely to experience landsliding. The best landslide identification approach that can aid in rapid response used all available pre-event SAR data from ascending and descending paths, topographic masks, and all available post-event data acquired within one week following the landslide triggering event. Landslide density "heat maps" help reduce noise and highlight landslides that can be used for rapid response. Importantly, we found that landslide detection capability increases rapidly over the first two months and then continues to increase slowly with more image acquisitions. This finding implies that

satellites with higher repeat acquisitions may provide more accurate landslide detection that can assist with rapid response. Alternatively, SAR data operating with longer radar wavelengths will help reduce noise and could improve landslide detection, especially for rapid response. Future SAR missions, like the L-band NASA-ISRO NiSAR mission, which is currently expected to launch in January 2022, will also provide publicly available data. If Google Earth Engine ingests the NiSAR data, our methodology could be used for Sentinel-1 and NISAR, which will undoubtedly improve the ability of those without SAR

expertise to utilize SAR-based amplitude change detection to monitor natural hazards.






**Appendix A: Additional figures showing ROC curves, SAR amplitude change maps for rapid response, and pre- and post-event SAR amplitude data.**



**Figure A1**: **ROC curves without and with the DEM mask**. Colored lines correspond to different pre-event ($T_{pre}$) and post-event ($T_{post}$) stack time periods. The legend shows a numbered index that corresponds to the Index number in Tables 1, S1, and S2. In general, the AUC scores are higher when a higher number of SAR images are included in each stack. The AUC scores are also higher for combined ascending + descending products than they are for separate ascending or descending tracks.



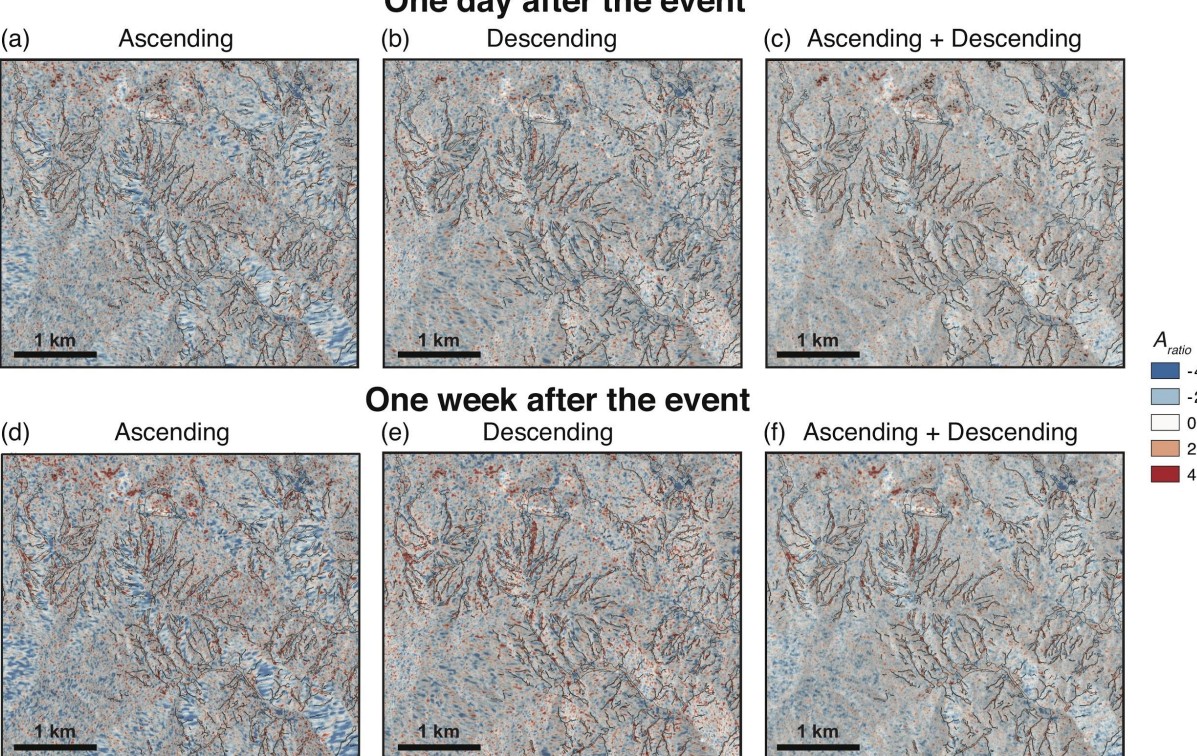

**Figure A2: SAR amplitude change maps for rapid landslide identification.** (a-c) SAR amplitude change using all pre-event SAR data and the first available post-event SAR data for ascending, descending, and combined ascending + descending. (d-f) SAR amplitude change using all pre-event SAR data and the available post-event SAR data acquired within one week

for ascending, descending, and combined ascending + descending. Black polygons show GSI landslide inventory.

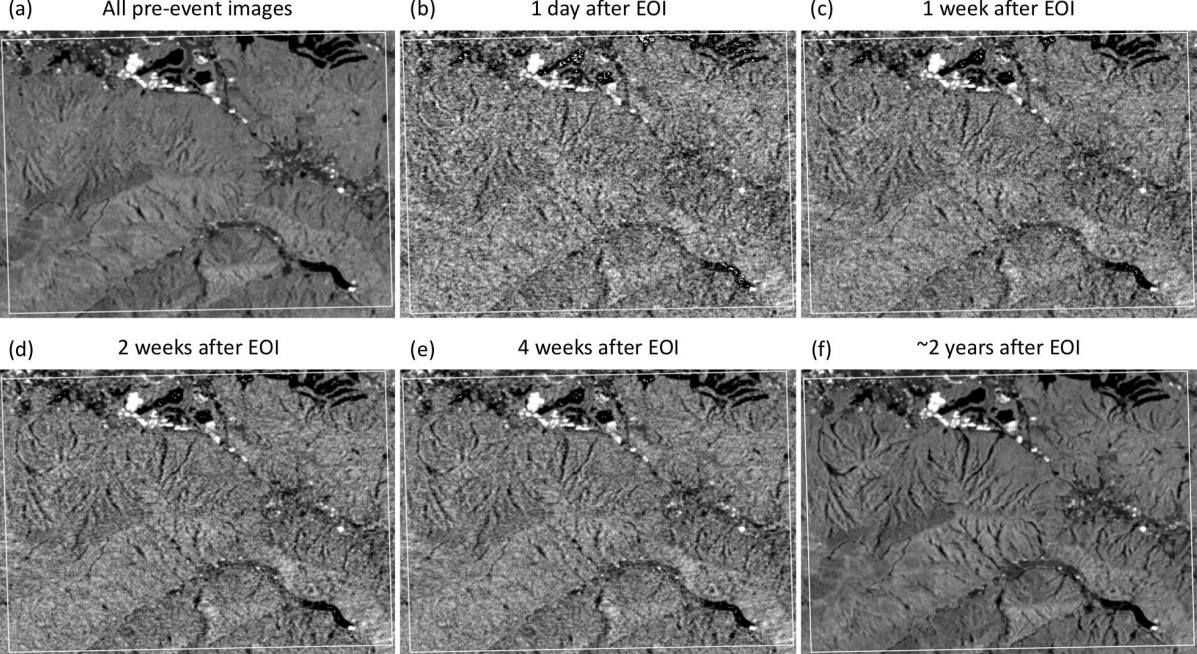

**Figure A3: SAR amplitude stacks.** (a) Stack of all available pre-event images. Stacks of post-event SAR data acquired up until (b) 1 day after the EOI, (c) 1 week after the EOI, (d) 2 weeks after the EOI, (e) 4 weeks after the EOI, and (f) 2 years after the EOI. The stacked images become gradually smoother with less noise as more data are acquired. Landslides appear with darker colors in the grayscale.

**Table S1**. List of all ascending SAR data used to make pre- and post-event stacks. Tabs show stack index numbers.

**Table S2**. List of all descending SAR data used to make pre- and post-event stacks. Tabs show stack index numbers.

**Data and Code availability.** The data used in this manuscript were provided by the National Aeronautics and Space Administration (NASA) and the European Space Agency (ESA) Copernicus program and accessed on Google Earth Engine (https://code.earthengine.google.com). Code for landslide identification on the Google Earth Engine is available at https://github.com/MongHanHuang/GEE_SAR_landslide_detection and https://doi.org/10.5281/zenodo.4060268. The Geospatial Information Authority of Japan (GSI) landslide inventory is available at https://www.gsi.go.jp/BOUSAI/H30.taihuu7gou.html.

**Author contribution**. ALH, SYJ, and MH designed the study and processed and analyzed the data. MH wrote the Google Earth Engine codes. ALH and MH performed the geomorphic analysis and landslide interpretation. PA and DBK helped select



the test site and provided advice on landslide mapping. PA provided the GSI landslide inventory. MH and HRK performed ROC analyses. ALH and MH wrote the manuscript with contributions from SYJ, PA, and DBK.

**Competing interests**. The authors declare no competing interests.


**Financial support.** Funding for this work came from NSF PREEVENTS-2023112 grant (ALH), NSF EAR-2026099 (MH), and High Mountain Asia NNX16AT79G and Disaster Risk Reduction and Response 18-DISASTER18-0022 (DBK and PA).

**Acknowledgements**. We thank the National Aeronautics and Space Administration (NASA), the European Space Agency
(ESA) Copernicus program, and the Google Earth Engine for providing freely available data and processing. We thank the Geospatial Information Authority of Japan for providing the landslide inventory https://www.gsi.go.jp/BOUSAI/H30.taihuu7gou.html). Part of this research was carried out at the Jet Propulsion Laboratory, California Institute of Technology, under a contract with the National Aeronautics and Space Administration (80NM0018D0004).

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
