# Peer review of "Rapid landslide identification using synthetic aperture radar amplitude change detection on the Google Earth Engine"

_Natural Hazards and Earth System Sciences, 2020_

## Referee Comment (RC1) · Anonymous Referee #1 · 26 Oct 2020

The Authors present a SAR amplitude change detection based tool to detect and map landslides events. The tool is implemented in Google Earth Engine and applied to Sentinel-1 imagery. The use of SAR amplitude for landslide detection has raised some interests recently, so as platforms for data pre-processing.

The manuscript is easy to read and 'pleasant' but somehow too vague in the description of some key concepts. in particular in the presentation of the algorithm (the tool can be easy to use, but the description of the variables and steps should be more detailed or rigorous).

Unfortunately, in my opinion, there is a main conceptual issue which does not allow

the Authors to obtain the result they want, and, as a matter of fact, invalidating the conclusions: the algorithm obtained the thresholds using an external inventory, without the inventory prepared by GSI, no thresholds, and no DEM mask. So the tool cannot be used to prepare inventories, without having already an inventory.

I then suggest to reject the paper for a methodological problem, and I encourage the Authors to review the research framework.

In detail:

Abstract

16 - triggering events because it penetrates clouds: probably referring to rainfall-induced events

16 - regularly acquired: this is true for S1, not sure it is true for other facilities.

25 – Not sure about the meaning of the entire sentence

1 Introduction

General question: are 'identify' and 'detection' synonyms? Since in the introduction also 'mapping' is mentioned, I suggest to define the terms, also because the first (wrong?) expectation I had was about a tool for mapping landslides (see row 35).

71-72 Our methodology. . .: actually it seems to me that you need to know that landslides occurred, and the changes were caused by landslides.

73 duration: also this point should be better clarified: how can I know a priori about the duration of the event? This can be 'simple' like in the test case, but how about landslides triggered by seasonal monsoons, or typhoons?

73 catastrophic: I suggest to cancel it.

2 Methods

92-93: it can be used for constructing landslide inventories with high accuracy, sounds

to me a bit vague. Does it mean that it produces high accuracy inventory or simply that it can contribute to? And, what does 'high accuracy' mean?

2.1 SAR amplitude-based Change Detection on Google Earth Engine

104 – 105 are processed to remove thermal noise and they are radiometric and terrain calibrated: sorry, unclear to me whether they are processed by your algorithm or that is the way images are available in GEE. In both cases, if radiometric calibrated, then it should not be amplitude but some backscatter coefficient, Sigma/beta/gamma_nought. Furthermore, what does 'terrain calibrated' mean? Orthorectified?

108 – 109 .. spatial resolution...: please check 25 x 25 (should be EW and not IW, but EW is not here used). Furthermore, I suggest you to explain why the two concepts (resolution and pixel spacing) are here both mentioned and what is the impact in the final maps of a spatial resolution of 20 x 22 m.

111 All S1 GRD amplitude values are provided in logarithm..: I guess this is something done during the pre-processing chain to stretch the values. And from here to the end, I think it is no more A but some backscatter coefficient (after radiometric correction)

116 a single landslide: again, the feeling I get is that we must already know that a landslide has occurred...

118 ... will vary..: I suggest to provide better indications on how to select T in case of rapid identifications or inventory constructions.

120 we further reduced noise: Why further? Is it referred to the multi-look done during the pre-processing level to get GRD images? Actually what is done here sounds to me a bit weird. The stack can smooth the signal, but I am not sure that it works as a filter to remove the noise (which is multiplicative). I think the median can depend on many factors including seasonal variations, soil moisture (rain..). Is this one of the reasons for which it is better to use many images? If yes, it is not just a matter of number but also a matter of the variability experienced in the study areas. Furthermore, I'm

surprised that the different geometries can be combined without problems, in particular in mountainous areas. Another question: the pixels in ascending and descending after multi-look should follow different Gamma distributions, how they can be combined using the mean?

124 examining the change in amplitude: really too generic. Examining how? I think we should not wait for paragraph 2.1.

132 using threshold-based masks: again too generic. How did you obtain thresholds? Are the thresholds general, or tuned according to the test areas? What data do you need to obtain the thresholds?

2.1 SAR Change Detection Performance

138 -139 To test the performance of our SAR..: I think this is one of the main issues of this paper. This is not a comparison, actually the inventory is used to tune the thresholds. So, no external inventory, no product. . .

141 -143 We calculated detection performance: I disagree, you are tuning the model, looking for the best threshold

154: the thresholds used for slope angle..: again, this is possible only if the external inventory is available.

3 Test site

4 Results

4.1 Landslide Identification

General question: can the need of having such a large number of images be originated by the way that images are smoothed?

180 we compared: I am strongly persuaded that you did not compare but used the GSI inventory to obtain the final map

185 136 images: but this method should produce a product in one or two weeks...

192 the complete stack: again, the product should be obtained just after the event.

215 from the previous analyses: this means including false positives, I guess. Indirectly, the GSI inventory is used too. Also, thresholds obtained seem to be very conservative...

220-221: this sentence does not have much sense. Empirically but without a landslide inventory... Need for verification.

222 and Fig.3 a: it says that, definitely, the best results cannot be obtained after one or two weeks.

4.2 Landslide identification for rapid response

232 the DEM mask: if I understood well, the DEM mask was obtained using the best result obtained in the previous analysis, which means by using the multitemporal stack. Please clarify.

271 .. removed isolated pixels: how was 6 chosen?

277 farmlands: this is a consequence of using such a large T-pre, isn't it?

5 Discussion

294-295: to be consistent with your analysis, I suggest to say an amplitude decrease, or better, surface backscatter.

314 helps to reduce the noise: again, I suggest to say that it smooth the signal. (which is different from reducing the noise, in particular, the multiplicative noise in SAR images).

318 inconsistent: sorry I can't understand the concept.

324 Gaussian smoothing filter: pay attention because, after multi-looking, pixels do not normally distribute.

[Figure]

372-373: this is to be removed, or results should be presented in more detail.

6 Conclusions

---

## Referee Comment (RC2) · Anonymous Referee #2 · 1 Nov 2020

The paper is well written and the presentation is clear.

However, I still have to suggest to reject the paper. The reason is that it misses certain elements of a scientific paper and is rather a technical report.

The main reason is that the results are never put into any context or prepared to any other method. So, we do not know if they are good or bad. Are other approaches better? It is nowhere shown.

The authors argue that because GEE makes the use of SAR data easy, so basically no other method can compare to that. That basic assumption of the paper may be correct, but even that could be questioned. Nevertheless, it is essential that the approach is

compared to other state-of-the-art methods, so that we can validate the approach and see if there are other methods that can get a higher accuracy. Then the authors could argue, that this is still acceptable (or not), because their approach is easier to use, doesn't require expensive software, etc. However, the lack of any comparison with other methods, makes it impossible to validate the importance and correctness of the work.

Furthermore, just dismissing coherence based methods remains also questionable. In the case of Sentinel-1, at least in areas with 6-day repeat cycle, coherence maybe acceptable. Again, the authors should prove that and it seems to me that this is an excuse, as GEE does not support this. Again, this leads back to the main point. There is no comparison to other methods.

---

## Author Comment (AC1) · 13 Nov 2020

**The author's response is shown below in blue text.**

**Anonymous Referee #2**

The paper is well written and the presentation is clear.

However, I still have to suggest to reject the paper. The reason is that it misses certain elements of a scientific paper and is rather a technical report.

The main reason is that the results are never put into any context or prepared to any other method. So, we do not know if they are good or bad. Are other approaches better? It is nowhere shown.

The authors argue that because GEE makes the use of SAR data easy, so basically no other method can compare to that. That basic assumption of the paper may be correct, but even that could be questioned. Nevertheless, it is essential that the approach is compared to other state-of-the-art methods, so that we can validate the approach and see if there are other methods that can get a higher accuracy. Then the authors could argue, that this is still acceptable (or not), because their approach is easier to use, doesn't require expensive software, etc. However, the lack of any comparison with other methods, makes it impossible to validate the importance and correctness of the work.

Furthermore, just dismissing coherence based methods remains also questionable. In the case of Sentinel-1, at least in areas with 6-day repeat cycle, coherence maybe acceptable. Again, the authors should prove that and it seems to me that this is an excuse, as GEE does not support this. Again, this leads back to the main point. There is no comparison to other methods.

We thank Referee #2 for their review of the manuscript. It is unfortunate they do not feel our manuscript contains all of the elements of a scientific paper. Nonetheless, we will address the issue they have raised.

The main issue Referee #2 has with our manuscript is that we did not directly compare our SAR change method to any other method. However, this is not the focus of our manuscript. We know from many previous studies that optical data provide the highest quality information for identifying landslides under cloud free conditions and we discuss the success of optical data in the Introduction section of our paper.

We do note that optical data from Sentinel-2, Landsat, and MODIS are currently available in Google Earth Engine (GEE) and therefore could be used to identify landslides. In fact, in our GEE codes that are available on Github, we include functionality for adding Sentinel-2 (Figure 2) and Landsat data, and we encourage those who are interested to develop methods using these data. But again, a detailed comparison between optical and SAR is beyond the scope of our manuscript as the main point of our work is to document a SAR based method that does not require downloading a large volume of data to a local system, or specialized processing software and training. Furthermore, we are not claiming it is the best possible method for identifying landslides, but rather it is one of several methods that can be used, particularly when there is significant cloud cover.

Lastly, Referee #2 would have liked us to have used Sentinel-1 coherence change to identify landslides. SAR coherence change has been used to successfully identify landslides but successful case studies are limited to urban areas or areas without dense vegetation (as is described in the Introduction and Discussion of our manuscript). However, these data are not available on Google Earth Engine. We do not feel that it would be useful to provide a comparison with SAR coherence since the users of our GEE codes would not be able to employ this approach. Furthermore, SAR-based coherence does not work well for identifying landslides in forested regions because coherence is always low and thus does not change (see Jung and Yun, 2019 for a detailed analysis).

Reference:
Jung, J. and Yun, S.-H.: Evaluation of Coherent and Incoherent Landslide Detection Methods Based on Synthetic Aperture Radar for Rapid Response: A Case Study for the 2018 Hokkaido Landslides, Remote Sens., 12(2), 265, https://doi.org/10.3390/rs12020265, 2020.

---

## Author Comment (AC2) · 13 Nov 2020

**The author's response is shown below in blue text.**

**Anonymous Referee #1**

The Authors present a SAR amplitude change detection based tool to detect and map landslides events. The tool is implemented in Google Earth Engine and applied to Sentinel-1 imagery. The use of SAR amplitude for landslide detection has raised some interests recently, so as platforms for data pre-processing.

The manuscript is easy to read and 'pleasant' but somehow too vague in the description of some key concepts. in particular in the presentation of the algorithm (the tool can be easy to use, but the description of the variables and steps should be more detailed or rigorous).

Unfortunately, in my opinion, there is a main conceptual issue which does not allow the Authors to obtain the result they want, and, as a matter of fact, invalidating the conclusions: the algorithm obtained the thresholds using an external inventory, without the inventory prepared by GSI, no thresholds, and no DEM mask. So the tool cannot be used to prepare inventories, without having already an inventory.

I then suggest to reject the paper for a methodological problem, and I encourage the Authors to review the research framework.

We thank the Referee #1 for their review of our manuscript. In the revised manuscript we will expand our description of the variables and steps embedded in our methodology (more details in the line by line comments below). Referee #1 raised some important points that will help improve our revised manuscript.

Referee #1 suggests that an external landslide inventory is required for our SAR-based methodology to detect landslides, but this is not the case. Similar to other types of remote sensing data, our SAR-based approach can be used to identify landslides regardless of any pre-existing landslide inventory because landslides cause a significant change in SAR backscatter intensity when comparing pre- and post-event SAR images. In fact, the main point of this method is to be able to identify landslides with no external landslide inventory. We only used an external inventory for two main reasons: 1) to simply show that our SAR change detection approach in GEE correctly identifies true landslides and 2) to objectively constrain the SAR backscatter change values in order to provide future studies with some rough guidelines for future landslide events. In the revised manuscript we will add text to make it clear that an external inventory is used as a "proof of concept" but is not required for landslide mapping.

We will address the more detailed comments below.

In detail:

Abstract
"16 - triggering events because it penetrates clouds: probably referring to rainfall- induced events

The ability for SAR to penetrate clouds is not only relevant to rainfall-induced landslides. For example, in the Introduction section we state "the 2015 Mw 7.8 Gorkha earthquake, which triggered more than 25,000 landslides, where persistent cloud cover prevented landslide mapping from satellite optical imagery for more than a week".

16 - regularly acquired: this is true for S1, not sure it is true for other facilities. 25 – Not sure about the meaning of the entire sentence

Our method is specifically designed for S1 data, which is currently the only SAR data available in Google Earth Engine.

1 Introduction

General question: are 'identify' and 'detection' synonyms? Since in the introduction also 'mapping' is mentioned, I suggest to define the terms, also because the first (wrong?) expectation I had was about a tool for mapping landslides (see row 35).

In our manuscript we used the words "identify" and "detect" as synonyms. We will remove the term "mapping" from the revised manuscript as we do not actually provide tools for the mapping. Our tools allow users to identify/detect landslides.

71-72 Our methodology. . .: actually it seems to me that you need to know that landslides occurred, and the changes were caused by landslides.

Our method can be used with or without any knowledge of landslides. However, we expect that future users of our methods are trying to locate landslides and have a good idea of where and when potential landslides have occured.

73 duration: also this point should be better clarified: how can I know a priori about the duration of the event? This can be 'simple' like in the test case, but how about landslides triggered by seasonal monsoons, or typhoons?

The duration of an event is user defined. We agree that in some cases this is simple, like an earthquake or a single storm. A typhoon is another good example where meteorological data is

available to help guide users. It does become more complicated when looking for landslides over entire seasons, for instance, triggered by seasonal monsoons. Nonetheless, a time period must be selected to compare pre- and post-event images. Our current focus is for identifying landslides triggered by distinct events with the time period clearly defined.

73 catastrophic: I suggest to cancel it.

We will remove the word "catastrophic" from the revised manuscript.

2 Methods

92-93: it can be used for constructing landslide inventories with high accuracy, sounds to me a bit vague. Does it mean that it produces high accuracy inventory or simply that it can contribute to? And, what does 'high accuracy' mean?

"High accuracy" refers to the ability to detect true landslides. To remove confusion, we will remove the words "high accuracy" from the revised manuscript.

2.1 SAR amplitude-based Change Detection on Google Earth Engine

104 – 105 are processed to remove thermal noise and they are radiometric and terrain calibrated: sorry, unclear to me whether they are processed by your algorithm or that is the way images are available in GEE. In both cases, if radiometric calibrated, then it should not be amplitude but some backscatter coefficient, Sigma/beta/gamma_nought. Furthermore, what does 'terrain calibrated' mean? Orthorectified?

All of these corrections/calibrations are made by GEE (https://developers.google.com/earth-engine/guides/sentinel1). The clarify this we will modify the text in the revised manuscript to read "GEE provides S1 Ground Range Detected (GRD) products that are pre-processed to remove thermal noise, have undergone radiometric calibration, and are orthorectified"

Referee #1 is correct that GEE provides the data as a backscatter coefficient in decibels ($10*\log_{10}\sigma°$). In the revised manuscript will better explain the GEE data format by adding more information to the text. We will add "All S1 GRD data values are provided in logarithmic units of decibels (dB) calculated as $10*\log10(\sigma°)$, where $\sigma°$ is the SAR backscatter coefficient. The backscatter coefficient is a measure that can be used to determine if the radar signal is scattered towards or away from the SAR sensor. When $\sigma° > 0$, the dominant scattering is toward the satellite. The direction of scattering is primarily controlled by the geometry of the landscape

relative and the electromagnetic properties of the land cover." We will also change "amplitude" to "backscatter coefficient" or "backscatter intensity" in the revised manuscript.

108 – 109 .. spatial resolution. . .: please check 25 x 25 (should be EW and not IW, but EW is not here used). Furthermore, I suggest you to explain why the two concepts (resolution and pixel spacing) are here both mentioned and what is the impact in the final maps of a spatial resolution of 20 x 22 m.

To remove confusion in the revised manuscript, we will modify the text and only list the available pixel spacing resolution provided by GEE. The revised text will read "GEE provides GRD images with 10, 25, or 40 m pixel spacing".

111 All S1 GRD amplitude values are provided in logarithm..: I guess this is something done during the pre-processing chain to stretch the values. And from here to the end, I think it is no more A but some backscatter coefficient (after radiometric correction)

This is correct and we will modify the text in the revised manuscript to describe that we are analyzing the SAR backscatter coefficient. All instances of "amplitude" will be changed to "backscatter coefficient" or "backscatter intensity".

116 a single landslide: again, the feeling I get is that we must already know that a landslide has occurred. . .

It is not a requirement to know that a landslide has occured to use our methodology, however we expect that people who are searching for landslides using our methods have suspected a landslide has occurred.

118 . . . will vary..: I suggest to provide better indications on how to select T in case of rapid identifications or inventory constructions.

We provide recommendations in the Results and Discussion sections. These recommendations are to use all of the available pre-event and post-event imagery (as also seen in Figs. 2, 3, 4). For rapid identification there will be a limited number of post-event imagery, while for landslide investigations that take place long after the event there can be hundreds of images.

120 we further reduced noise: Why further? Is it referred to the multi-look done during the pre-processing level to get GRD images? Actually what is done here sounds to me a bit weird. The stack can smooth the signal, but I am not sure that it works as a filter to remove the noise (which is multiplicative). I think the median can depend on many factors including seasonal variations, soil moisture (rain..). Is this one of the reasons for which it is better to use many

images? If yes, it is not just a matter of number but also a matter of the variability experienced in the study areas. Furthermore, I'm surprised that the different geometries can be combined without problems, in particular in mountainous areas. Another question: the pixels in ascending and descending after multi-look should follow different Gamma distributions, how they can be combined using the mean?

Stacking SAR images reduces transient error and noise from sources like atmospheric effects or short term changes in soil moisture. Referee #1 is correct that seasonal changes are important to consider and we describe that throughout the manuscript. That is why we tested a wide range of $T_{pre}$ and $T_{post}$. To further clarify, we will add some more text to the revised manuscript that reads "We also reduce error and noise from atmospheric delay and other sources by stacking images to create pre-event and post-event stacks. SAR data stacking has been shown to significantly improve the signal-to-noise ratio in SAR data (e.g., Cavalié et al., 2008; Zebker et al., 1997)."

The combination of ascending and descending SAR data is one of the major findings from our study. By combining the ascending and descending stacks we found an improvement in the landslide detection success, both visually (Figure 2d-f) and as is shown by the ROC analyses (Figure 3c).

124 examining the change in amplitude: really too generic. Examining how? I think we should not wait for paragraph 2.1.

We will add more detailed description early on in the revised manuscript. The revised manuscript will read "We analyzed changes in SAR backscatter intensity data from the Copernicus Sentinel-1 (S1) satellite constellation to identify landslides. Landslides can cause a significant increase or decrease in the backscatter intensity depending on how landslides alter the hillslope geometry and dielectric properties of the land cover (Adriano et al., 2020; Jung and Yun, 2019)". Also in the revised manuscript we added an expanded description of the SAR backscatter coefficient that reads "The backscatter coefficient is a measure that can be used to determine if the radar signal is scattered towards or away from the SAR sensor. When $\sigma° > 0$, the dominant scattering is toward the satellite. The direction of scattering is primarily controlled by the geometry of the landscape relative to the radar look direction and the electromagnetic properties of the land cover."

132 using threshold-based masks: again too generic. How did you obtain thresholds? Are the thresholds general, or tuned according to the test areas? What data do you need to obtain the thresholds?

In general, landslides are not initiated in flat areas and or at the hilltop. Therefore for landslide identification it is expected to be an improvement to remove flat zones and hilltops. However, the true distribution of slopes affected by landslides will be site dependent so we suggest users explore a range of slope and curvature thresholds. We will add more information to the revised manuscript in the Methods section. It will read "Since landslides often occur on steep hillslopes, slopes less than a few degrees can be masked, as these are the areas that generally correspond to non-landslide locations. Adriano et al. (2020) suggests that slopes > 12° are susceptible to landslides, however these slope thresholds can vary in different regions. Additionally, it is common for landslides to runout into lower slope areas, therefore it is important to initially consider a wider range of slope values when searching for landslide deposits. We will explore the threshold values in detail in the Results section by analyzing the GSI landslide inventory, however we encourage future investigations without an external landslide inventory to explore a wide distribution of slope and curvature thresholds in order to help improve their landslide identification."

2.1 SAR Change Detection Performance

138 -139 To test the performance of our SAR..: I think this is one of the main issues of this paper. This is not a comparison, actually the inventory is used to tune the thresholds. So, no external inventory, no product. . .

We disagree with Referee #1's interpretation here.. As mentioned above, an external inventory is not necessary to identify landslides. We only use an external inventory to validate our method and to quantify the SAR backscatter intensity thresholds that correspond to true landslides for this case study. The goal of this analysis is to provide recommendations for future studies that do not have an external inventory. Without the use of an external inventory, we would still be able to identify landslides based on our interpretation of SAR change detection. As with any remote sensing data set, a secondary dataset for validation is useful to confirm true landslides. Nonetheless it is clear from the Reviewer's comments that we did not clearly describe our approach and we will make numerous changes to the revised manuscript to make sure this is clearly described.

We will modify this paragraph to read "To test the performance of our SAR intensity change detection for landslides, we compared our results to the inventory provided by the GSI for our ~277 km$^2$ AOI. This comparison was performed to show that our SAR intensity change methods in GEE can be used to identify true landslides. We note that in many cases, especially for rapid response, an external landslide inventory will not exist prior to investigation, which will make it more challenging to quantify the SAR change identification performance. The main goal of our SAR change identification performance analyses is to quantify the SAR intensity change values that correspond to true landslides in order to provide useful guidelines for future investigations

that only have SAR data available. It is important to emphasize that our GEE-based SAR change identification approach can be used to identify landslides without an external landslide inventory."

141 -143 We calculated detection performance: I disagree, you are tuning the model, looking for the best threshold

Here we are exploring the range of SAR change values that maximize the true detection. This step is required to document the utility of our new methodology, it is not required for our method to be used to detect landslides and is only intended to help guide future investigations that do not have an external landslide inventory.

154: the thresholds used for slope angle..: again, this is possible only if the external inventory is available.

It is still possible to use slope thresholds without an external inventory. We know, without an external inventory, that landslides are unlikely to occur in flat areas or on hilltops, and it is thus possible to mask out these areas. We again, only use the external inventory to determine the best parameters for our case study that can help guide future studies. We will modify the revised manuscript to better describe these issues and will add a reference to Stanley, T., & Kirschbaum, D. B. (2017). A heuristic approach to global landslide susceptibility mapping. Natural hazards, 87(1), 145-164.

3 Test site

4 Results

4.1 Landslide Identification

General question: can the need of having such a large number of images be originated by the way that images are smoothed?

We found that the large number of images is needed to improve the signal-to-noise ratio and reduce error in individual SAR images.

180 we compared: I am strongly persuaded that you did not compare but used the GSI inventory to obtain the final map

Again, the GSI inventory is not needed to identify landslides. It is used to confirm to the readers that our SAR change detection is capable of detecting true landslides. It is also used to help us

objectively provide a range of SAR intensity change values that correspond to landslides to help guide future investigations. Please see Figure 2 and Figure A2. These figures show the SAR backscatter change data, there is no "tuning".

185 136 images: but this method should produce a product in one or two weeks. . .

This is not the rapid response section. This section is about landslide initiation using all available data at the time of study.

192 the complete stack: again, the product should be obtained just after the event.

This section is about landslide identification overall, not for rapid response. Rapid response is discussed in Section 4.2.

215 from the previous analyses: this means including false positives, I guess. Indirectly, the GSI inventory is used too. Also, thresholds obtained seem to be very conservative. . .

We agree the DEM mask thresholds are conservative and do not remove much of the terrain, but this is the best result to maximize the AUC.  This wide range of slope values is because the landslides are deposited in some lower slope areas. Figure 3 shows a wide range slope and curvature thresholds can be used with slight changes in AUC.

220-221: this sentence does not have much sense. Empirically but without a landslide inventory. . . Need for verification.

We will modify the revised manuscript to read "Based on our objective AUC analyses, we recommend that these same thresholds can be used as starting points for landslide identification in future events in different regions, but note that thresholds will likely need to be adjusted since the distribution of slope and curvature values for landslides will vary depending on the region."

222 and Fig.3 a: it says that, definitely, the best results cannot be obtained after one or two weeks.

That is correct. We found better results over the 277 km$^2$ AOI a much longer time period after the EOI. This statement is not meant for the Rapid Response section.

4.2 Landslide identification for rapid response

232 the DEM mask: if I understood well, the DEM mask was obtained using the best result obtained in the previous analysis, which means by using the multitemporal stack. Please clarify.

That is correct. We will modify the revised manuscript to read "the DEM mask from the previous analysis in Section 4.1"

271 .. removed isolated pixels: how was 6 chosen?

We will modify the revised text to read "We also removed locations that we attributed to noise by masking out isolated pixels that had less than 6 neighboring apparent "landslide" pixels"

277 farmlands: this is a consequence of using such a large T-pre, isn't it?

That may be true, but we did not explore the SAR change in farmlands in sufficient detail to clarify this statement.

5 Discussion
294-295: to be consistent with your analysis, I suggest to say an amplitude decrease, or better, surface backscatter.

We will modify the revised manuscript to read " This decrease in SAR backscatter intensity occurs because the landslide scar and damage acts to decrease backscattering reflectance to the satellite relative to a pre-failure ground surface (Adriano et al., 2020; Jung and Yun, 2020). However, there were some landslides or parts of landslides that also had negative $I_{ratio}$ values. Therefore, for future investigations we recommend exploring both positive and negative $I_{ratio}$ values to search for landslide features."

314 helps to reduce the noise: again, I suggest to say that it smooth the signal. (which is different from reducing the noise, in particular, the multiplicative noise in SAR images).

We will delete this paragraph from the revised manuscript.

318 inconsistent: sorry I can't understand the concept.

We will delete this paragraph from the revised manuscript.

324 Gaussian smoothing filter: pay attention because, after multi-looking, pixels do not normally distribute.

We will delete this paragraph from the revised manuscript.

372-373: this is to be removed, or results should be presented in more detail.

6 Conclusions

---

## Editor Comment (EC1) · Mahdi Motagh (Editor) · 17 Nov 2020

Dear authors, I asked the reviewer to have a look at your reply letter. Below you will see his/her reply.

————

I'd like to thank the Authors the for the answers.

One of my main concerns was related to the use of external data to calibrate the model. The Authors 'clarify' that these data are actually used 1) to show that the approach (not at all described, then) correctly identifies true landslides, and 2) to constrain the SAR

backscatter changes value for future studies.

Unfortunately I'm not fully persuaded about the answers because to me (absolutely personal opinion...) Paragraph 2.1 and Paragraph 4.1 seem to say something different. Also assuming that these data are not used to calibrate the model, then, quite a few sentences would loose meaning:

'Receiver Operating Characteristic (ROC) and Area Under the Curve (AUC) analyses to determine the best approach for landslide identification'

'The quality of the SAR-based landslide detection depends on the total number of images used in the pre-event and post-event stacks, slope and curvature threshold, a ratio threshold, and whether we use ascending, descending, or combined ascending + descending data'

and so on...

In the end, it seems to me that the link between the external info and the procedure is too strong to be resolved simply saying that the real purpose is to constrain the signal for future studies, and I don't see the way (and the sense) to untie the bonds in the current paper structure.

Since there are some potentialities in the paper, what I actually suggest is to re-define the framework, propose the method independently from the 'sensitivity analysis', with a more robust validation which uses the external data without entering in any calibration/sensitivity procedure.

I'm sorry to say that I keep my opinion and I suggest to reject the paper.

---

## Editor Comment (EC2) · Mahdi Motagh (Editor) · 17 Nov 2020

Dear authors, I asked the reviewer to evaluate your reply letter. Below you will see his/her reply:

——— actually, I don't think it is convincing. Let me reply to the authors below: I understand the authors point of view, but I don't think they got mine. So, what the authors should ask themselves, what is the novelty of the paper and what is the value for the reader? Currently, the paper is a technical report basically stating that it is possible to detect landslides with GEE. But there is no further information given. Is it a good detection method? How does it compare to other methods?

[Figure]

So, what is the benefit for the reader? He can learn that it is possible to use GEE to detect landslides. I would assume nobody was seriously doubting that. The question one would have would be, I can use GEE, but do I get good results and how do these results compare to other methods? What are the advantages and disadvantages compared to other well-known methods?

So, the paper remains a work report, where the authors report what they did, but it does not provide valuable guidance for the readers. This is just another way of describing my initial review.

---

## Author Comment (AC3) · 27 Nov 2020

We thank Referee #2 for their response. As is mentioned in our previous response, and throughout our manuscript, the novelty of our paper is that we document, for the first time, a way to use the cloud-based Google Earth Engine (GEE) to detect landslides and we show a clear relationship between the number of pre- and post-event SAR images and detection success. To our knowledge, there is no previously published manuscript documenting the use of SAR amplitude data to identify landslides in Google Earth Engine and therefore it is not fair to assume whether or not people would doubt its feasibility. Furthermore, our work highlights the importance of rapid SAR image

collections and how the accuracy of our landslide detection technique increases with the number of images acquired following a landslide event.

We attempted to clearly outline the main advantages/disadvantages of our method throughout our manuscript but will make sure to further clarify these points in our revised manuscript. The main advantage of using SAR, as numerous previous studies have documented, is that SAR is able to "see" through clouds and therefore we may have information on the location of landslides for rapid response when cloud cover is preventing the use of optical data. Being able to do this in GEE, which requires no specialized software or data storage, will enable many more people to search for landslides using SAR data. The main disadvantages are related to the limitations based on the satellite and landform geometry (which is also an issue for optical) and noise and error in the Sentinel-1 images.

While it is beyond the scope of our manuscript to make direct comparison of our method with other landslide detection methods, by making direct comparison with an external inventory we are effectively comparing to an optical-based inventory that has a high accuracy. Our AUC analyses provide an objective comparison with the external landslide inventory and we find AUC ranging from $\sim$0.6 for rapid response up to 0.8 with many post-event SAR images. In the revised manuscript we will clarify the main advantages and disadvantages of our new approach to put our findings in better context.

---

## Author Comment (AC4) · 27 Nov 2020

We thank Referee #1 for their response. We are confident that we can thoroughly address their concerns during our revisions. In particular, we will make sure to better describe our approach and results in the absence of an external inventory. We will add a separate section that describes our results without the use of the external inventory and a section that will use ROC analyses to perform sensitivity tests. Thanks again for your time and constructive feedback.